# Respiration Rates, Metabolic Demands and Feeding of Ephyrae and Young Medusae of the Rhizostome *Rhopilema nomadica*

**Zafrir Kuplik** [1,2], **Dani Kerem** [1] and **Dror L. Angel** [1,*]

1   Department of Maritime Civilizations, Charney School of Marine Sciences & Recanati Institute for Maritime Studies, University of Haifa, Mt. Carmel, Haifa 31905, Israel; zafrirkuplik@tauex.tau.ac.il (Z.K.); dankerem@research.haifa.ac.il (D.K.)
2   Steinhardt Museum of Natural History, Tel Aviv University, Tel Aviv 69978, Israel
*   Correspondence: dangel@univ.haifa.ac.il

**Abstract:** Jellyfish (cnidarians and ctenophores) affect the marine food web through high feeding rates and feeding efficiency, but in contrast to their great importance in the ecosystem, our knowledge of their dietary requirements is limited. Here we present the results of respiratory and feeding trials of the rhizostome *Rhopilema nomadica*, the dominant scyphozoan in the waters of the Eastern Mediterranean, which often establishes massive swarms, mainly in the summer months. Through multiple measurements of oxygen demand in *R. nomadica* at bell diameters of 3–49 mm, we were able to assess its minimum energetic requirements. These, and the results of the feeding trials on individuals of the same bell diameter range, show that *R. nomadica* is a very efficient predator. When presented with prey concentrations of 100 prey items per liter, a single hourly feeding session provided between 1.15 and 3 times the estimated daily basal carbon requirement. Our findings suggest that *R. nomadica* is well adapted to its environment, the hyperoligotrophic waters of the eastern Mediterranean, able to efficiently exploit patches of plankton, possibly at rates even higher than what we observed under laboratory conditions.

**Keywords:** scyphomedusa; Eastern Mediterranean; minimum energetic demand; predation potential; feeding specialization

## 1. Introduction

Most scientific inquiries into the predatory impact of gelatinous zooplankton—such as scyphomedusae, hydromedusae and ctenophores—on plankton communities have focused on investigations of jellyfish diets. These studies have generally involved gut content analysis of jellyfish that were sampled in situ and laboratory studies in which digestion times and clearance rates are determined [1,2]. Although they are informative, these methods rely on assumptions that may lead to inaccurate conclusions (reviewed by Purcell [3]). For example, confinement effects and excessive prey density in laboratory feeding experiments could lead to underestimated or overestimated feeding and clearance rates, respectively, whereas gut content analysis of field-sampled specimens is spatiotemporally dependent; since plankton distribution in the oceans is generally not homogeneous but patchy (e.g., Haury et al. [4]). Moreover, diet composition and the number of prey items revealed through the analysis of gut contents of free-swimming animals may show high variability, even between adjacent sampling sites, sampled concurrently [5]. Similarly, the diet of jellyfish varies greatly due to diel and seasonal differences in the distribution of their preferred prey [6]. An alternative approach, which also relies on certain assumptions, are laboratory measurements of respiration and feeding rates. These enable us to estimate the predator's minimum energetic demand and, in turn, to estimate the minimum amount of prey required to sustain it.

Since the mid-1980s, less than a decade after it was first recorded, the rhizostome *Rhopilema nomadica* has demonstrated a remarkable ability to proliferate and form massive

seasonal swarms on a yearly basis [5] in a region known for its ultra-oligotrophy [7]. In view of this, our general working hypothesis was that it should possess relatively high metabolic efficiency and predation potential. This could be tested by evaluating several widely used metabolic measures, also targeted by this study:

1.  Basal respiration rate—oxygen consumption (measured) and $CO_2$ production (estimated). The prediction for these measures in *R. nomadica* is to be relatively low.
2.  Basal/minimum carbon demand, needed to cover the estimated $CO_2$ production, accordingly also predicted to be low.
3.  Ingestion rate (measured) and carbon input (estimated), predicted to be high when compared to similarly sized individuals of other species.
4.  Carbon budget—input versus required, when evaluated in terms of required feeding duration to satisfy daily demands, predicted to be relatively low in *R. nomadica.*

In addition, we attempted to correlate different biometric features of the animals. Since respiration rates are commonly associated with biomass of the medusa, either wet/dry weight or carbon content [8,9], and since mass correlates with the bell diameter (BD) [10] it is suggested that only one of these measures is needed to estimate the others. Inferring both carbon content and respiration rates from BD will enable us to study the organisms in situ and in the laboratory with minimal disruption and related artifacts. We will attempt to specifically address the above predictions when discussing our results.

## 2. Materials and Methods

All *R. nomadica* medusae used for the experiments were cultured from polyps in the marine laboratory of the Faculty of Marine Sciences, Ruppin Academic Center, in Michmoret, Israel. The medusae were reared in kreisel tanks [11], supplied with flowing 50 μm-filtered sea water at ambient temperature and fed with newly hatched *Artemia* sp. nauplii twice a day. Medusae with similar bell diameters were selected for each trial. In order to avoid stressing the medusae prior to the experiments, BD was first estimated by eye and was measured only after the experiments. For the small ephyrae, BD was uniformly assumed to be 3 mm. A starvation period of 10–15 h was applied in order to ensure there was no food present in the guts of the medusae (i.e., that all prior eaten food had been digested) at the beginning of the respiration and feeding experiments. During the experiments, temperature and salinity in the laboratory flowing seawater system ranged from 21 to 29 °C and from 39.5 to 40 psu, respectively.

### 2.1. Respiration Experiments

Experimental design details are presented in Table 1. Oxygen consumption measurements were carried out in screw-top glass jars ("respiration chambers") of 0.02 L, 0.4 L and 1.0 L, (jar volume was determined to match size of the medusa), filled with 50 μm of aerated, pre-filtered seawater and submerged in a flow-through water table (i.e., measurements were performed at ambient temperature). Each respiration chamber contained one or two medusae, except for ephyrae, where each chamber contained 10 individuals. The control jars (one control in each trial) contained only filtered sea water. In order to avoid bubbles, the 0.02 L vials were slowly over-filled and then sealed with parafilm prior to closing the lid; larger jars were sealed and closed underwater in a large tank. After inserting medusae into the respiration chambers, they were observed to ensure they behaved normally, i.e., swam at a normal rate as was observed in the rearing tank. At the end of the respiration measurements, the experimental medusae were placed on millimeter paper to measure their BD and then briefly rinsed with distilled water before they were freeze-dried. After tissues were freeze-dried (VirTis Benchtop K lyophilizer), dry weight (DW) was measured using a Precisa XT 220 A analytical balance. Organic carbon (C) was analyzed from the lyophilized tissue using an Elementar Vario Micro Cube at a combustion temperature of 1150 °C. Precision of the method was 3%. The analysis was performed at the Marine Biology Station of the National Institute of Biology, Piran, Slovenia.

**Table 1.** Respiration experimental design. $N$ = number of respiration chambers, $n$ = number of medusae in each respiration chamber, $V$ = volume of each respiration chamber, $t$ = duration of respiration measurement. BD = bell diameter $\pm$ standard deviation.

| Date | Water Temperature (°C) | $N$ | $n$ | BD (mm) | $V$ (L) | $t$ (min) |
|---|---|---|---|---|---|---|
| 3 September 2014 | 29.8 | 3 | 10 | 3 | 0.02 | 180 |
| 10 September 2014 | 29 | 3 | 1 | $14 \pm 1$ | 0.02 | 120 |
| 14 September 2014 | 29 | 3 | 1 | $15 \pm 1$ | 0.02 | 130 |
| 1 October 2014 | 27.5 | 3 | 1 | $8 \pm 0$ | 0.02 | 180 |
| 15 October 2014 | 26.5 | 3 | 1 | $34 \pm 1$ | 1 | 380 |
| 23 October 2014 | 26 | 3 | 1 | $49 \pm 3$ | 1 | 230 |
| 19 November 2014 | 23 | 3 | 1 | $39 \pm 1$ | 0.4 | 130 |
| 2 December 2014 | 21 | 3 | 2 | $24 \pm 1$ | 0.4 | 130 |

*2.2. Respiration Measuring Equipment and Procedure*

Oxygen consumption was measured using a 4-channel fiber-optic oxygen meter (FireSting $O_2$, PyroScience GmbH, Regensburg, Germany) via light sensitive sensor spots glued to the inner wall of the respiration chambers. The sensor spots were calibrated at 0 and 100% oxygen saturation shortly before the beginning of the respiration experiments. While measuring, the fiber-optic exciting light sources were attached to the outer side of the respiration chambers using designated adaptors (PyroScience Basic Spot Adaptor), opposite to the sensor spots. A submersible temperature sensor (TSUB36, Temperature sensor for FireSting $O_2$, Teflon-coated and submersible, shielded cable, ca. 3.6 mm, 2 m cable) was placed in the water table in order to continuously adjust the solubility coefficient of oxygen for variations in the ambient temperature. In addition, in order to avoid possible calibration drift, the sensors were calibrated twice during the 2-month experimental period. Preliminary trials showed a lag of approximately 15–20 min until oxygen readings stabilized, therefore, only data recorded after this time lag were analyzed. This time lag also allowed the medusae to acclimatize to the respiration chambers.

Oxygen concentration readings were recorded every minute during the measurements. In order to eliminate the possibility of differences in oxygen consumption due to natural diurnal behavior of the medusae (i.e., differences in activity levels at different hours of the day), all respiration trials were performed in the evening hours. Furthermore, all incubation chambers were covered with an opaque, light-proof plastic sheet in order to avoid extreme changes in light intensity which might change the activity levels of the medusae. The average dissolved oxygen concentration in the respiration containers at the end of the experiments never dropped below 75% of the initial concentration.

*2.3. Regression Analyses of Respiration Rates*

Oxygen concentration for each respiration trial was plotted against time (duration of the respiration trial), after subtracting the background control measurements from the experimental measurements, and expressed by a linear regression line. Respiration rate (RR) was calculated as:

$$\text{RR} \left( \mu\text{mol } O_2 \text{ ind}^{-1}\text{min}^{-1} \right) = b \times \left( V \times n^{-1} \right)$$

where $b$ is the slope (mol min$^{-1}$) of the regression, $V$ is the volume of the incubation chamber and $n$ is the number of medusae in the chamber. Respiration rates were expressed as mL $O_2$ medusa$^{-1}$ d$^{-1}$, using the following conversion factor: 1 μmol $O_2$ = 0.022391 mL $O_2$ [12]. Respiration rates were plotted against DW and C of the medusae. In addition, mass-specific respiration rates were calculated (i.e., mL $O_2$ mg DW$^{-1}$ d$^{-1}$ or mL $O_2$ mg C$^{-1}$ d$^{-1}$). The plotted biometric relationship (DW = 0.008 $\times$ BD$^{2.7075}$ and C = 0.0016 $\times$ BD$^{2.3176}$), derived from specimens with a range of BD (Figure 1), enabled mass estimation for medusae the biomasses of which were not recorded.

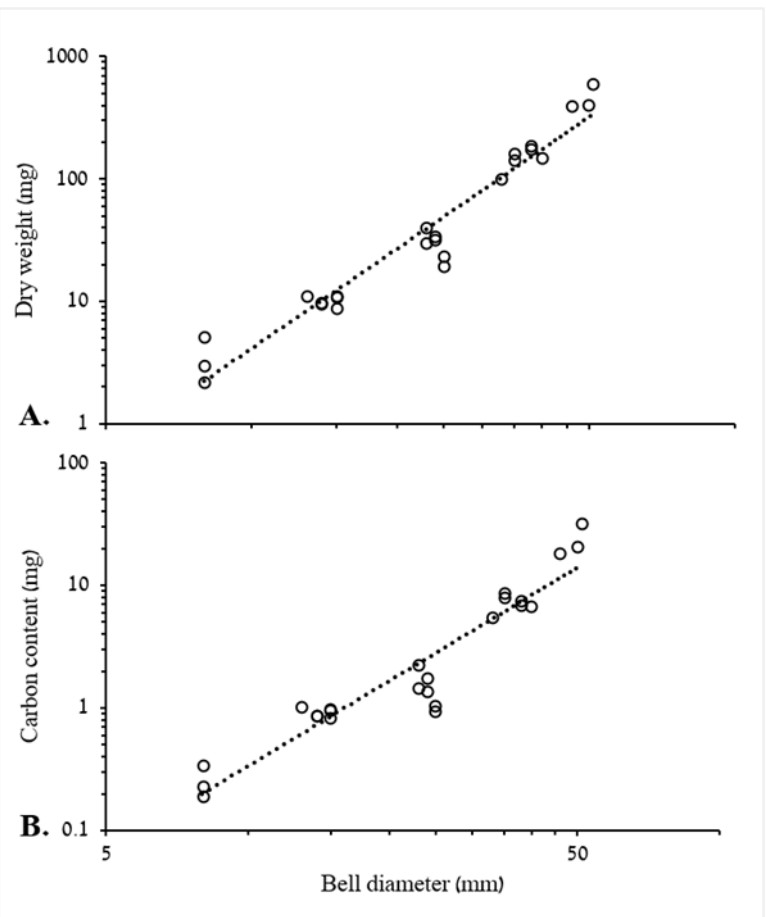

**Figure 1.** Log–log plots of the relationship between bell diameter of *R. nomadica* with (**A**) dry weight (DW) and (**B**) carbon content (C) ($n$ = 24). (**A**) DW = 0.008 × BD$^{2.7075}$, R$^2$ = 0.943. $F_{1,22}$ = 363.8, $p < 0.001$. (**B**) C = 0.0016 × BD$^{2.3176}$, R$^2$ = 0.896. $F_{1,22}$ = 188.5, $p < 0.001$.

### 2.4. Minimum Daily Carbon Demand

The amount of carbon lost through respiration was estimated by using a respiratory quotient $\frac{CO_2}{O_2}$ of 0.8. This value is assumed to sufficiently describe the molecular ratios between $CO_2$ expelled and $O_2$ consumed for various zooplanktonic carnivores, including gelatinous zooplankton [13,14]. By multiplying the calculated number of $CO_2$ moles expelled by the molar mass of carbon (12.01 g mole$^{-1}$), minimum daily carbon demand values were estimated (mg C ind$^{-1}$ d$^{-1}$). These values were later compared with the number of prey consumed by *R. nomadica* in the feeding experiments and the carbon content of the prey. This also enabled calculation of weight-specific minimum daily carbon demand ratio (% d$^{-1}$) by dividing the minimum daily carbon demand (mg C ind$^{-1}$ d$^{-1}$) by C of the medusa and multiplying it by 100.

### 2.5. Feeding Experiments

Ingestion and clearance rate for individual *R. nomadica*, fed on newly hatched *Artemia* sp. nauplii, were measured under laboratory conditions in feeding chambers of 2, 17 and 20 L (Table 2).

**Table 2.** Feeding trial setups: Newly hatched *Artemia* sp. nauplii were used as prey in all feeding trials. *N* = number of experimental tanks, *V* = volume of tank, *C* = concentration of prey, *t* = duration of each feeding trial. BD = bell diameter ± standard deviation.

| Date | Temperature (°C) | N | BD (mm) | V (L) | C (*Artemia* $L^{-1}$) | t (min) |
|---|---|---|---|---|---|---|
| 28 September 2014 | 28 | 3 | 15 ± 1 | 2 | 100 | 60 |
| 1 October 2014 | 28 | 4 | 10 ± 1 | 2 | 97 | 60 |
| 9 October 2014 | 28 | 3 | 16 ± 1 | 2 | 100 | 30 |
| 13 October 2014 | 27.5 | 3 | 35 ± 1 | 20 | 99 | 60 |
| 18 October 2014 | 27 | 4 | 9 ± 1 | 2 | 50 | 60 |
| 18 October 2014 | 27 | 4 | 9 ± 1 | 2 | 50 | 30 |
| 20 October 2014 | 25.5 | 5 | 50 ± 1 | 20 | 50 | 60 |
| 24 October 2014 | 25 | 5 | 21 ± 1 | 2 | 50 | 60 |
| 31 October 2014 | 24.5 | 6 | 16 ± 1 | 2 | 74 | 60 |
| 1 November 2014 | 24.2 | 4 | 42 ± 4 | 17 | 53 | 45 |
| 18 November 2014 | 23 | 5 | 40 ± 2 | 17 | 62 | 60 |

In order to minimize the confinement effect, the small feeding chambers (2 L) were used for medusae ≤21 mm, while chambers of 17 and 20 L were used for larger medusae. All feeding trials were conducted at ambient water temperature in dark conditions; the experimental set-up was covered with a light-proof plastic sheet in order to prevent *Artemia* sp. nauplii, which have positive phototaxis, from congregating in an uneven manner in the feeding containers. Three control containers were used in each feeding trial and contained only prey (*Artemia* sp. nauplii) organisms. In order to increase the probability of homogeneous distribution of prey in the experimental containers, feeding trials that were conducted in the 2 L containers were performed on a bottle roller (Rolacell RC 42, New Brunswick Scientific, Edison, NJ, USA) at a rotation speed of 60 RPH. The room temperature in which the bottle roller was located was adjusted to ambient water temperature. The 17 and 20 L containers were placed in a water table at ambient water temperature (Table 2). Medusae were acclimated in the experimental containers for 30 min before adding the prey and were observed for normal pulsation activity before starting the feeding trials. At the end of the experiments, each medusa was gently transferred from its experimental container, using a ladle, into a small glass bowl where its BD was measured, as described above. In order to minimize the chance of removing prey that was not ingested, only enough water to cover the medusa in the ladle was used. After measurement, the medusae were returned to the kreisels from which they were taken for the feeding trials. Prey used in the 2 L containers were added individually until reaching the desired number of prey (see below), whereas 3–5 sub-counts out of the *Artemia* culture were used to determine the volume of prey culture required to achieve the desired initial prey concentrations in the 17 and 20 L experimental containers. The remaining prey were concentrated using a 100 μm mesh and fixed by adding Lugol's acid solution (2% final concentration). The number of remaining prey was counted the following day using a Motic SMZ-171 stereoscope.

Prey abundances of 50–100 individuals $L^{-1}$, commonly used in feeding experiments of jellyfish [15,16], were used in the feeding experiments. These prey concentrations were chosen since they enabled us to obtain significant predation readings and in a short period of time. Using lower prey density could result in a complete removal/predation of prey by the medusae before the end of the feeding trial, which would prevent the estimation of the medusae feeding rate (i.e., predation per unit of time). The relatively short duration of the feeding trials was preferable since it minimizes the effect of factors which could compromise the validity of the results, e.g., stress to the medusae due to confinement and varying encounter rates with the prey due to a decrease in the activity and abundance of prey.

### 2.6. Predation Rates

Feeding Trials

Ingestion rate ($I$, prey medusa$^{-1}$ h$^{-1}$) was calculated using the equation:

$$I = \frac{N_{tc} - N_{te}}{t}$$

where $N_{tc}$ and $N_{te}$ are the remaining number of prey in the control and experimental feeding containers, respectively, at the end of the feeding trial. $t$ is the duration (h) of the feeding trial.

Clearance rate ($F$, L medusa$^{-1}$ h$^{-1}$) was calculated using the equation [17]:

$$F = \left(\frac{V}{n \times t}\right) \times \ln\left(\frac{N_{tc}}{N_{te}}\right)$$

where $V$ is the volume of the experimental container and $n$ is number of predators in each container.

### 2.7. Conversion Factors

In order to compare mass-specific respiration rates with values published in the literature, DW: wet weight ratio of 4.87% was used, based upon a mean ratio obtained from in situ samples of *R. nomadica* ($n$ = 7) [5], and the relationship between measured DW and carbon content was plotted (Figure 2). Biometric conversion factors were obtained from the respiration trials data regressions and used to predict mass and respiration rates of the medusae in the laboratory feeding experiments DW = 0.008 × BD$^{2.7075}$ (Figure 1A), C = 0.0016 × BD$^{2.3176}$ (Figure 1B), RR = 0.0005 × BD$^{2.2873}$ (Figure 3). *Artemia* carbon content was determined as a rough average of several published values (Table 3): 1 *Artemia* sp. nauplius = 1 µg C.

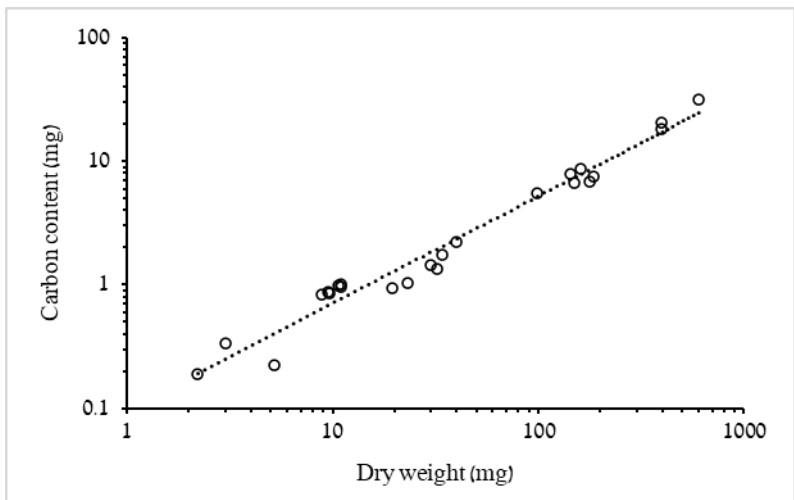

**Figure 2.** Log–log plots of the relationship between carbon content (C) and total dry weight (DW) of *R. nomadica* ($n$ = 24). C = 0.0973 × DW$^{0.8642}$, R$^2$ = 0.969. $F_{1,22}$ = 685.7, $p < 0.001$.

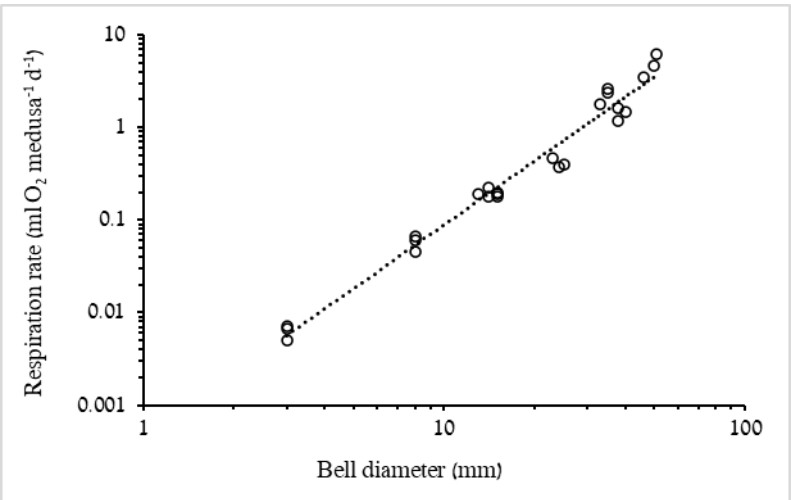

**Figure 3.** Log–log plot of respiration rate of *R. nomadica* as a function of BD ($n = 24$). RR = 0.0005 × BD$^{2.2873}$, R$^2$ = 0.9764. $F_{1,22} = 874.1$, $p < 0.001$.

**Table 3.** *Artemia* spp. carbon content as reported in published studies.

| *Artemia* spp. | Carbon Content (µg) of Newly Hatched Nauplii | Reference |
|---|---|---|
| *A. salina* | 0.5 | [18] |
| *A. salina* | 0.3 | [19] |
| *A. franciscana* | 0.93 | [20] |
| *A.* sp. | 1 | [21] |
| *A.* sp. | 0.69–1.63 | [22] |

*2.8. Statistical Analysis*

Since temperature varied between respiration trials (21–30 °C), we tested for its effect on the measured respiration rates by a multiple linear regression, with log-transformed respiration rates as the dependent variable and log-transformed DW and temperature as the independent covariables. Regression analysis was used to estimate the relationships among variables in both respiration and feeding experiments. Significance of the regression models was determined using ANOVA and a significant fit was determined when $p < 0.05$. SPSS Volume 21 software was used for statistical analyses.

**3. Results**

*3.1. Respiration Trials*

3.1.1. Biometric Relationships of Bell Diameter and Mass

Of the 54 *R. nomadica* used in the respiration trials, 24 medusae were measured for BD and mass (DW and C). The BD of the largest medusa was 51 mm, with a DW of 602 mg, while the smallest medusa had a BD of 8 mm and a DW of 2.2 mg. Log–log plots show that 94% and roughly 90% of the variation in DW and carbon content could be accounted for by BD, respectively (Figure 1). These relationships were later used for biomass estimations of ephyrae and medusae in the feeding experiments, in which only BD was measured.

Carbon content correlated positively with dry weight of the medusae (Figure 2). Nevertheless, the allometric exponent *b* (=0.8642, 95% confidence interval: 0.795–0.933) indicates a less than isometric increase of carbon content with mass. On average, medusae heavier than 29 mg dry weight had about half the mean C:DW ratio (%) shown by individuals <29 mg and only 40% of the mean C:DW ratio (%) estimated for ephyrae (Table 4).

**Table 4.** Summary of the data used in the analyses. $N$ = total number of individuals in each trial, DW = dry weight, C = carbon content, RR = respiration rate, DW-RR= dry-weight specific respiration rate, C-RR = carbon-specific respiration rate. [†] Only RRs were measured for these individuals. Values in bold, for ephyrae, were estimated by using regression equations of the rates calculated from other medusae. [††] The presented RRs are half the measured RRs ($n$ = 2 in each experimental chamber).

| $N$ | BD (mm) | Dry Weight (mg) | Carbon Content (mg) | C%/DW | RR (mL $O_2$ ind$^{-1}$ d$^{-1}$) | DW-RR (mL $O_2$ mg DW$^{-1}$ d$^{-1}$) | C-RR (mL $O_2$ mg C$^{-1}$ d$^{-1}$) | Carbon Demand (%) of Body C d$^{-1}$ |
|---|---|---|---|---|---|---|---|---|
| 30 [†] | 3 | 0.21 ± 0.04 | 0.03 ± 0.00 | 12.2 ± 0.3 | 0.006 ± 0.001 | 0.030 ± 0.001 | 0.249 ± 0.001 | 10.7 ± 0.0 |
| 3 | 8 | 3.47 ± 1.55 | 0.25 ± 0.08 | 8.0 ± 3.5 | 0.057 ± 0.011 | 0.018 ± 0.004 | 0.238 ± 0.058 | 10.2 ± 2.4 |
| 3 | 14 ± 1 | 10.47 ± 0.84 | 0.94 ± 0.07 | 9.0 ± 0.3 | 0.183 ± 0.006 | 0.018 ± 0.001 | 0.194 ± 0.012 | 8.3 ± 0.5 |
| 3 | 15 ± 1 | 9.73 ± 0.95 | 0.89 ± 0.08 | 9.2 ± 0.3 | 0.203 ± 0.016 | 0.021 ± 0.002 | 0.229 ± 0.032 | 9.8 ± 1.4 |
| 6 [††] | 24 ± 1 | 29.78 ± 7.45 | 1.46 ± 0.48 | 4.8 ± 0.5 | 0.413 ± 0.045 | 0.014 ± 0.004 | 0.301 ± 0.093 | 12.9 ± 4.0 |
| 3 | 34 ± 1 | 134 ± 31.48 | 7.37 ± 1.64 | 5.5 ± 0.1 | 2.252 ± 0.401 | 0.017 ± 0.001 | 0.308 ± 0.018 | 13.2 ± 0.8 |
| 3 | 39 ± 1 | 170.67 ± 18.15 | 7.01 ± 0.40 | 4.1 ± 0.3 | 1.437 ± 0.237 | 0.008 ± 0.002 | 0.205 ± 0.029 | 8.8 ± 1.3 |
| 3 | 49 ± 3 | 465 ± 118.65 | 23.64 ± 7.24 | 5.0 ± 0.4 | 4.760 ± 1.305 | 0.010 ± 0.001 | 0.203 ± 0.021 | 8.7 ± 0.9 |

### 3.1.2. Respiration Rate and Carbon Demand

Multiple linear regression of log-transformed respiration rates data against log-transformed medusa dry weights and temperature, showed that temperature within the range tested in this study did not significantly affect respiration rates ($p$ = 0.07), however, medusa biomass did have an effect ($F_{1,19}$ = 692.5, $p < 0.001$). Therefore, temperature was eliminated from further analyses.

Oxygen consumption increased significantly ($p < 0.05$) with both BD and biomass (Figures 3–5). However, whereas exponent $b$ of respiration rate as a function of dry weight was 0.8575 (95% confidence interval: 0.789–0.926), showing a negative allometric relationship in a narrow range of 0.01–0.02 mL $O_2$ mg DW$^{-1}$ d$^{-1}$ (Figure 4), the slope of respiration rate as a function of carbon mass ($b$ = 0.9862, 95% confidence intervals 0.915–1.058) was almost isometric, showing a slight decrease in carbon-specific respiration rates throughout the tested range of medusa weights in this study (Figure 5). Estimated mass-specific respiration rates calculated for ephyrae support these observations as well (Table 4).

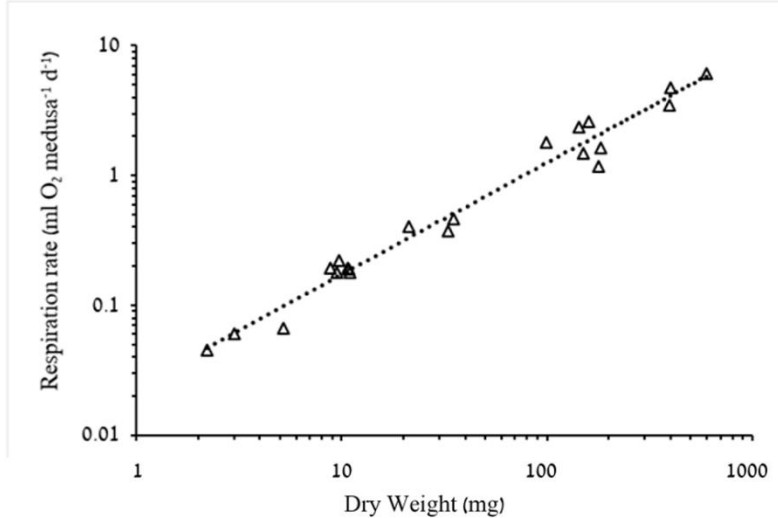

**Figure 4.** Log–log plots of the relationship between dry weight (DW) and respiration rate (RR) ($n$ = 21). RR = 0.0241 × DW$^{0.8575}$, $R^2$ = 0.9732. $F_{1,19}$ = 712.3, $p < 0.001$.

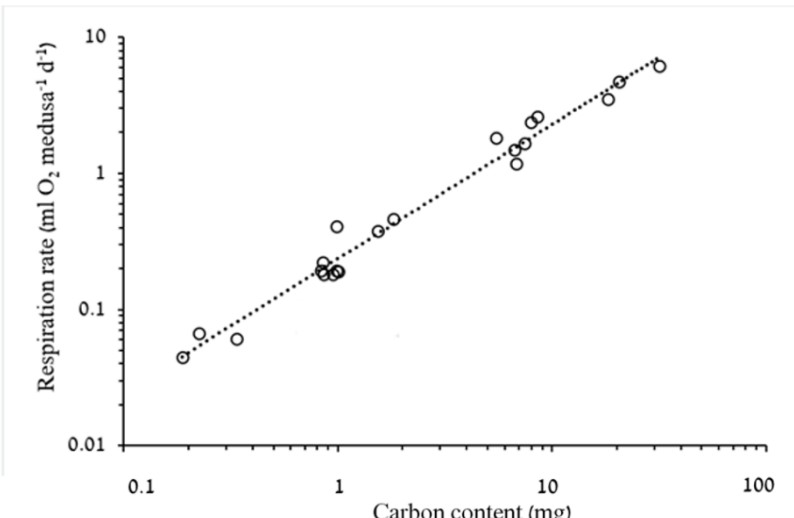

**Figure 5.** Log–log plots of the relationship between carbon content (C) and respiration rate (RR) ($n = 21$). RR = $0.2364 \times C^{0.9862}$, $R^2 = 0.978$. $F_{1,19} = 827$, $p < 0.001$.

Mean daily carbon demand for the medusae (assuming respiratory quotient = 0.8) was 10.3% ($\pm2.5\%$) of C (Table 4), i.e., approximately 10% of the total body carbon must be replaced on a daily rate, to meet the carbon lost as $CO_2$ during respiration. Estimated daily carbon demand calculated for ephyrae was similar (10.7%).

### 3.2. Feeding Experiments

#### 3.2.1. Ingestion and Clearance Rates

The two plotted curve lines for trials in which 50 and 100 prey $L^{-1}$ were used (35 medusae were tested) (Figure 6), show a significant exponential increase in ingestion rate against BD (Figure 6A). It is also apparent that prey abundance affected ingestion rate, with an almost linear relationship; ingestion rate for the lower abundance (i.e., 50 *Artemia* $L^{-1}$) is approximately half of the ingestion rate calculated for the highest prey abundance, 100 *Artemia* $L^{-1}$, at same BD size. A significant exponential increase against BD is apparent also for clearance rate (Figure 6B). Irrespective of prey density, curve lines almost overlap each other throughout the entire tested size range of the medusae, with the largest difference at the smallest BD; clearance rate for an 8 mm *R. nomadica* at 100 *Artemia* $L^{-1}$ is only 19% higher than clearance rate at 50 *Artemia* $L^{-1}$.

#### 3.2.2. Weight-Specific Ingestion Rate

The negative slopes of carbon-specific ingestion rate (C-specific *I*, mg $C_{(prey)}$ mg C medusa$^{-1}$ h$^{-1}$) as a function of BD at both prey concentrations (Figure 7) indicate that smaller medusae ingested more carbon, in relation to their body carbon content, than did larger medusae.

While not studied systematically, the ingestion rates measured during the single 45 min and the two 30 min feeding sessions (Table 2) were only slightly higher than during 1 h feeding sessions at the same prey concentrations. Therefore, 1 h sessions were used uniformly as the unit feeding time in order to assess the daily carbon budget, i.e., how many 1 h sessions are needed to repay the daily basal carbon loss.

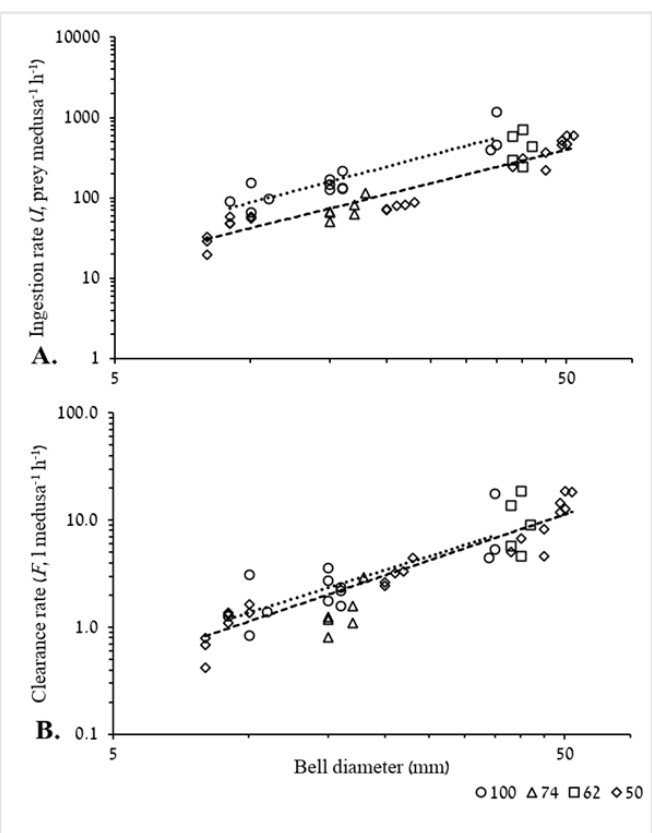

**Figure 6.** Log–log plot of ingestion rate (I) (**A**) and clearance rate (**B**) as a function of BD at different prey concentrations of 50, 62, 74 and 100 Artemia $L^{-1}$ ($n = 46$). Relatively similar prey concentrations are presented under the same legend values as follows: $50_{(n = 22)}$ = 50 and 53 Artemia $L^{-1}$, $100_{(n = 13)}$ = 97, 99 and 100 Artemia $L^{-1}$. Dashed curve line for 50 prey $L^{-1}$ and dotted curve line for 100 prey $L^{-1}$. (**A**) I (50 prey $L^{-1}$) = 1.705 × $BD^{1.39}$, $R^2$ = 0.907, $F_{1,20}$ = 194.2, $p < 0.001$. I (100 prey $L^{-1}$) = 2.924 × $BD^{1.48}$, $R^2$ = 0.813, $F_{1,11}$ = 47.8, $p < 0.001$. (**B**) F (50 prey $L^{-1}$) = 0.043 × $BD^{1.43}$, $R^2$ = 0.913, $F_{1,20}$ = 204, $p < 0.001$. F (100 prey $L^{-1}$) = 0.066 × $BD^{1.32}$, $R^2$ = 0.649, $F_{1,11}$ = 20, $p < 0.05$.

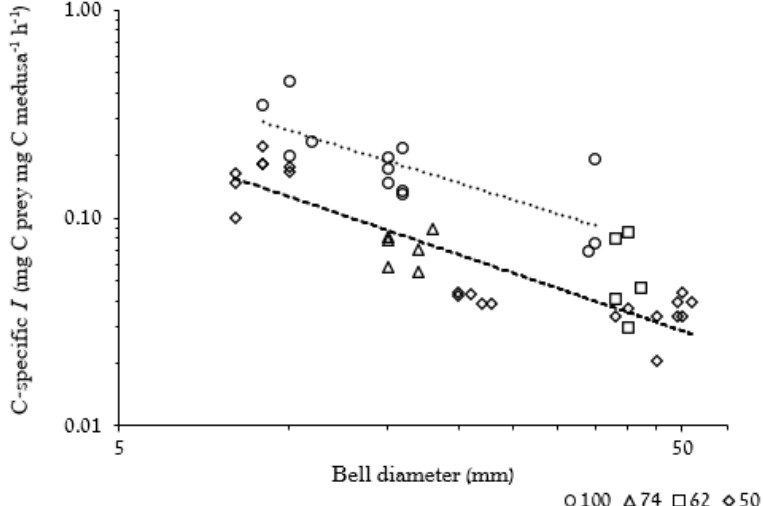

**Figure 7.** Log–log plot of carbon-specific ingestion rate (C-specific *I*) against BD ($n = 46$). Similar prey concentrations are presented under the same legend values as follows: $50_{(n = 22)}$ = 50 and 53 *Artemia* $L^{-1}$, $100_{(n = 13)}$ = 97, 99 and 100 *Artemia* $L^{-1}$. Dashed curve line for 50 prey $L^{-1}$ and dotted curve line for 100 prey $L^{-1}$. C-specific *I* (50 prey $L^{-1}$) = 1.065 × $BD^{-0.92}$, $R^2$ = 0.811, $F_{1,20}$ = 89.8, $p < 0.001$. C-sp *I* (100 prey $L^{-1}$) = 1.828 × $BD^{-0.84}$, $R^2$ = 0.584, $F_{1,11}$ = 15.1, $p < 0.05$.

Table 5 summarizes the data pertaining to the daily carbon budget, as derived from the results of the respiratory and feeding trials. They are grouped according to prey concentration. It appears that the minimum daily carbon requirement may readily be paid through a cumulative period ranging between 20–192 min of foraging (0.33–3.2 1 h sessions), at prey concentrations as used in our experiments. Prey concentration is limiting also in this respect, since at low and medium prey concentrations, >1 hourly sessions are required (except for the smallest BD), while at high prey concentrations <1 session is sufficient.

**Table 5.** Feeding trial data.

| N | BD (mm) | Carbon Content (mg) | Prey Concentration (ind L$^{-1}$) | Minimum Prey Requirement (ind d$^{-1}$) | *I* (ind Medusa$^{-1}$ h$^{-1}$) | *F* (L Medusa$^{-1}$ h$^{-1}$) | Required Number of 1 h Feeding Sessions |
|---|---|---|---|---|---|---|---|
| 8 | 9 ± 1 | 0.255 ± 0.060 | Low | 26 | 44 ± 15.5 | 1.1 ± 0.4 | 0.60 |
| 5 | 21 ± 1 | 1.906 ± 0.274 | Low | 191 | 78 ± 7.2 | 3.2 ± 0.8 | 2.45 |
| 5 | 40 ± 2 | 8.089 ± 0.798 | Low | 809 | 451 ± 192 | 10.3 ± 5.9 | 1.79 |
| 4 | 42 ± 4 | 9.326 ± 1.804 | Low | 933 | 286 ± 66.9 | 6.2 ± 1.7 | 3.26 |
| 5 | 50 ± 1 | 13.867 ± 0.797 | Low | 1387 | 528 ± 72.8 | 15.2 ± 3.1 | 2.62 |
| 3 | 16 ± 1 | 0.942 ± 0.079 | Med | 94 | 74 ± 22.4 | 1.6 ± 1.0 | 1.24 |
| 4 | 10 ± 1 | 0.335 ± 0.063 | High | 34 | 102 ± 36.2 | 2.8 ± 0.7 | 0.33 |
| 3 | 15 ± 1 | 0.897 ± 0.079 | High | 90 | 149 ± 17 | 1.5 ± 0.8 | 0.60 |
| 6 | 16 ± 1 | 1.021 ± 0.195 | High | 102 | 157 ± 51.2 | 1.9 ± 0.4 | 0.65 |
| 3 | 35 ± 1 | 5.931 ± 0.227 | High | 593 | 674 ± 425 | 9.1 ± 7.4 | 0.87 |

Means (±SD) of: bell diameter (BD), carbon content (C) (as estimated from BD by use of the equation C = 0.0016 × BD$^{2.3176}$ derived in the respiration trials, Figure 1B), ingestion rate (*I*) and clearance rate (*F*). Minimum prey requirement was derived using a mean basal metabolic rate of 10% C d$^{-1}$ (Table 3) and a prey C of 1 µg ind$^{-1}$ (Table 3). The required number of 1-h feeding sessions is arrived at by dividing values in column 5 by those in column 6. Prey concentration (ind L$^{-1}$): Low = 50–62; Med = 74; High = 97–100.

## 4. Discussion

### 4.1. Biometric Relationships

Our biometric characterization of 3–51 mm *R. nomadica*, being the first to date, showed robust BD-to-mass relationships (Figures 1 and 2, Table 4), similar to other studied species, enabling their subsequent use to estimate the biomass of individuals for which it was not obtained. The proportion of C in the dry biomass is often reported in metabolic studies and the range recorded for *R. nomadica* in the present study, 5–8%, corresponds well with values reported for other scyphomedusae, e.g., *Aurelia aurita* (4.3%, 3.7%) ([23,24], respectively), *Chrysaora fuscescens* (7.7%) [25] and *Rhizostoma pulmo* (5.6%, for juvenile medusa) [14]. Nevertheless, several other scyphozoan species showed higher ratios: 12.8% for *Cyanea capillata* [23], 11.1% for *Chrysaora quinquecirrha* [26] and 12% for *Phyllorhiza punctata* [27], to name a few. Collating numerous datasets, Lucas et al. [28] reported that on average, percentages differ between orders (i.e., Rhizostomeae, Semaeostomeae and Coronatae), hence, comparison and use of conversion factors should be limited to taxonomically affiliated species. It is noteworthy that dry weight varies with ambient salinity changes [29,30], thus comparing C/DW ratios between environments of different salinities should take this into account.

Our results suggest that this percentage differs not only between orders, but may also be size-dependent within the same species (i.e., *R. nomadica*); with smaller medusae having higher values (an average of 8.7% for medusae ≤16 mm and 4.8% for medusae >23 mm (Table 4)) and the regression relating %C to DW having an exponent of 0.86 (Figure 2). Higher ratios of total organics (lipids, proteins and carbohydrates) to DW in smaller individuals were also reported for *Aurelia aurita* [31], where total organics of medusae <20 mm were roughly two times higher than the ratios found in larger medusae (22.76% and 12.26%, respectively).

An additional conversion factor that was used in this study to estimate metabolic demand of medusae for which oxygen consumption was not measured (i.e., medusae of feeding experiments), was the prediction of respiration rate using a regression equation of respiration as a function of the medusa's BD (Figure 3).

*4.2. Respiration Rates*

The close association between the respiration rate of an organism and its biomass (W) is best expressed by the power function RR = a × W$^b$, where a is a normalization constant and *b* is the allometric scaling exponent [32,33]. Furthermore, *b* (slope) in this function had been assumed to be 0.75 in what is known as the "3/4 power law" [34,35], i.e., small animals have higher specific metabolic rates than large animals [36]. The universality of this law, however, has been challenged (e.g., [37,38]) and large differences in metabolic scaling were shown to be a prevailing trait during the ontogeny of many pelagic organisms [39], including gelatinous zooplankton where *b* was found to have a value between 0.8 and 1 [8,14]. Possible explanation for isometric scaling in jellyfish may lie in their continuous and exponential growth for most of their lives, if sufficiently fed [40].

Our results are in good agreement with these observations, where mass-dependent coefficients (*b*) were 0.86, for DW (Figure 4) and 0.99, for C (Figure 5). The latter finding is to be expected since C scaled on DW with a *b* of 0.86 (Figure 2). Larson [9] argued that mass-specific respiration rates of ephyrae and young medusae should be higher than those of larger medusae, since respiration rates are positively correlated with pulsation rates [41] and pulsation rates are negatively correlated with BD [42]. Indeed, higher specific respiration rates of ephyrae and young medusae than those of larger *A. aurita* were reported by Kinoshita et al. [43] (dry weight-specific RRs) and Moller and Riisgard [44] (carbon-specific RRs).

As regards the magnitude of the negative allometry, whereas our results show a 60% difference in DW-specific RRs between the two medusae size groups (≤15 mm and >15 mm), in *A. aurita* ephyrae of similar mass as the medusae used in the present study both DW-specific RRs [43] and carbon-specific RRs [44] were 5 times higher than those of adult *A. aurita* medusae.

Dry weight is often used to standardize gelatinous zooplankton metabolic rates; however, it appears that carbon content is more suitable for this purpose [45,46]. In general, the carbon-specific respiration rates obtained in the present study were within the range of rates recorded for other scyphomedusae, including semeaostomes and rhizostomes (0.05–0.5 mL $O_2$ mg $C^{-1}$ $d^{-1}$, reviewed by Larson [8]. However, since respiration rates of rhizostomes are usually higher than those of semeaostomes [14], possibly due to their enhanced swimming activity [47,48] comparison between species within the same order should be more representative.

A comparison of our results to other taxonomically related species (Table 6) indicates that, on average, carbon-specific respiration rates of *R. nomadica* were higher than published values for other rhizostomes. This is in contrast to our prediction and we suggest three possible explanations: (1) Different conversion factors, (2) temperature effects and (3) size of the experimental individuals used. The lack of reliable conversion factors for many rhizostomes is a potential source of error in metabolic studies. We used our own conversion factors for all species compared in Table 6, however, although all compared species belong to the family Rhizostomatidae, conversion factors may vary substantially. Uye [49] assumed, for example, that the carbon content of *Nemopilema nomurai* was 0.13% of wet weight, similar to the ratio found for *A. aurita* [24]. According to this C%/WW ratio, carbon-specific RRs of *N. nomurai* should be in the same range as *R. nomadica* (0.22), despite the considerable differences in size.

**Table 6.** Carbon-specific respiration rates of *R. nomadica* and other Rhizostomatidae species. * Carbon-specific respiration rate = mL $O_2$ mg $C^{-1}$ $d^{-1}$. ± standard deviation. Published mass of similar species was converted to carbon using conversion factors found for *R. nomadica*: DW%/WW = 4.87 [5], C% of DW = 4.85 (present study, when medusa dry weight > 11 mg).

| Species | Carbon Mass (mg) | Temperature (°C) | Carbon-Specific Respiration Rate * | Reference |
|---|---|---|---|---|
| *Rhopilema nomadica* | 0.25–23.64 | 21–29 | 0.24 ± 0.06 | Present study |
| *Rhizostoma pulmo* | 36.3 | 16, 20 | 0.17 | [50,51] |
| *Rhizostoma octopus* | 100, $1 \times 10^6$ | 15 | 0.16 | [52] |
| *Nemopilema nomurai* | $1.89 \times 10^3$–$1.89 \times 10^4$ | not specified | 0.12 | [49] |

Although in the present study temperature effect on respiration rate was insignificant, most of the metabolic trials in this study were conducted at ambient water temperatures, higher than 25 °C; almost 10 °C higher than the temperatures in the compared studies listed in Table 6. It is therefore possible that the relatively lower specific metabolic rates observed for them are due to temperature differences ($Q_{10} \leq 2$). Purcell [3] showed that the average $Q_{10}$ of *Aurelia* spp. across the ambient temperature range of 10–30 °C was 1.67. A similar $Q_{10}$ ratio was also observed for *Chrysaora quinquecirrha* in Chesapeake Bay [53]. On the other hand, Dawson and Martin [54] found that there was no difference between metabolic rates of tropical and temperate *A. aurita* spp. at very different ambient temperatures. Other metabolic studies which were conducted in laboratory conditions at manipulated temperatures, showed a much greater variety in $Q_{10}$ values (~3.2) (reviewed in Purcell [3]). Due to these large differences between laboratory manipulated temperatures and ambient temperatures, Purcell [3] argued that manipulated conditions lead to biased results and measurements should be conducted at the natural (ambient) range of temperatures for each species.

Even though the three species used for comparison (Table 6) are at different size scales, ranging from 36 to $1 \times 10^6$ mg C, they show similar carbon-specific respiration rates. Except for *Rhizostoma pulmo*, which was in the same order of magnitude size-wise, our animals were considerably smaller, likely with a higher metabolic rate.

In addition, several other factors should be taken into consideration when attempting to compare between studies or to use metabolic rates of starved organisms which were obtained in relatively small respiratory chambers and extrapolate these rates to medusae in-situ. Confinement effect and reduced swimming activity due to volume/mass (mL/g) ratios lower than 50 were shown to significantly reduce respiration rates [3,55]. Since these ratios in the present study ranged between 93 and 500, the metabolic rates reported here are believed to be accurate.

### 4.3. Carbon Demand

Metabolic rates are a useful tool for estimating the minimum nutritional requirements of an animal in order for it to sustain its metabolic costs. Here we used the conservative respiratory quotient (RQ) of 0.8 to evaluate *R. nomadica*'s minimal daily carbon budget. On average, the daily amount of carbon required to support the basal metabolic demand of starved *R. nomadica* at the ambient temperature range of 21–29 °C was 10.3% (±2.5) of its total body carbon content. It is noteworthy that the amount of carbon the medusa needs to ingest is greater than this amount, since carbon assimilation efficiency is estimated at 80% [56]. Thus, the carbon intake should exceed the above calculated value of 10.3% by at least 25% (i.e., 12.8% of total body carbon content) in order for the medusa to compensate for its metabolic loss. In general, a daily carbon turnover of 10.3% is slightly higher than the daily carbon demand values estimated for other medusae [8] which were mostly lower than 10%, once more in contrast to our prediction. However, as mentioned above, the higher temperatures at which we conducted our study, contrasts with the temperatures

at which the other studies were conducted (3–22 °C), and this may explain that difference. This assumption is supported by the findings of Ishii and Tanaka [57], where they estimated a 9.45% daily carbon demand of *A. aurita* at 24 °C.

### 4.4. Feeding Experiments

The purpose of the feeding experiments in this study was twofold: (1) Obtain for the first-time quantitative data on *R. nomadica*'s predation potential under controlled laboratory conditions and (2) compare daily carbon ration with basal metabolic requirements.

#### 4.4.1. Ingestion and Clearance Rates

Although diet composition and feeding behavior of several rhizostomes were previously described [58,59], to the best of our knowledge published data on feeding experiments of rhizostomes on *Artemia* sp. under laboratory conditions are missing. In fact, the only other study that enabled comparison was that of Morand et al. [60] on the semaeostome *Pelagia noctiluca*. We compared the mean ingestion and clearance rates of five *R. nomadica* individuals with a mean BD of 21 ± 1 to a point on the composite curve in Figure 3B of Morand et al. [60], constructed for *P. noctiluca* of 20 mm BD. In both studies, prey consisted of *A. salina* nauplii; our trials were conducted at 25.5 °C and those of Morand's at 21 °C. Since the curve relates ingestion rate (and derived clearance rate) to the geometric mean of the initial and final prey concentrations (presumed by the authors to represent the mean concentration during the trial) we computed this value for our five individuals, arriving at a mean of 25 prey $L^{-1}$. The ingestion rate for that point on the curve and the mean rate of our five individuals are 15 and 78 prey $medusa^{-1} h^{-1}$, respectively, roughly a 1:5 ratio, in agreement with our prediction. The corresponding hourly clearance rates are 0.65 and 3.6 $L h^{-1}$, respectively. The ingestion rates in both studies, being prey-density dependent, most probably underestimate the rate that would have been observed under constant prey concentration conditions [61].

In accordance with the study of Moller and Riisgard [16] on *Aurelia aurita* and other studies mentioned therein on that species and on the hydromedusa *Sarsia tubulosa*, within our studied prey concentration range, the ingestion rate increased linearly with prey concentration, with constant clearance rate. These authors therefore suggested that: " ... jellyfish may be well adapted to feed in patches of food, always being able to exploit their clearance capacity". This statement may even better befit rhizostome medusae, which are thought to originate in the oligotrophic waters of the tropics [62] and may have evolved to become patch-feeding specialists. Apart from their extensive prey-capturing apparatus, the strong swimming behavior, a common feature of rhizostomes [47,63], which results in high marginal flow velocities [48], allow rhizostomes in general and *R. nomadica* in particular to entrain more prey in its oral arms than the relatively slow "cruising" semaestomes. Actually, such features may also serve the medusae, when traveling between patches, to trap some prey at very low concentrations, prey that would have evaded other predators.

In the natural environment of *R. nomadica*, prey concentrations derived from net tows are generally much lower [64] than those used in the feeding trials. However, the measured feeding rates may approximate or even underestimate their grazing potential when they encounter a rare dense patch of plankton [65]. Since rhizostomes share similar physiological and predatory features [55,66,67] inter-order comparison of the measured predation rates should enable us to better evaluate *R. nomadica*'s predation potential. D'Ambra et al. [48] suggested that the high marginal flow velocities measured for *Phyllorhiza punctata* should enable it, and other rhizostomes, to efficiently capture rapidly swimming prey. On the other hand, Larson [55] showed that the rhizostome *Stomolophus meleagris* selected for slower zooplankton, suggesting that fast-swimming prey, such as calanoid and cyclopoid copepods, could sense the swimming medusa through water movement [68] and escape. Despite these contrasting conclusions, it seems clear that because rhizostomes have no central mouth, the size of prey they can ingest is restricted by the diameter of the mouthlet

openings covering the branched oral arms of rhizostomes [55,64,67] to mainly small plankton. It is important to note that although frequently used as prey in feeding experiments of gelatinous zooplankton, *Artemia* are not natural marine prey and are relatively slow swimmers in comparison to many copepod species [48]. Hence, the observed predation rates may not accurately reflect predation rates for other plankton species.

### 4.4.2. Carbon Budget

Within the tested size range, for *R. nomadica* individuals presented with a prey concentration of 100 prey L$^{-1}$, a single hourly feeding session provided between 1.15 and 3 times the estimated daily basal carbon requirement (Table 5). It is noteworthy that our respiration measurements were conducted in the absence of prey and that metabolic rates in feeding medusae should be higher. Indeed, Moller and Riisgard [44] showed that while feeding, *A. aurita* ephyrae and medusae had much higher respiratory rates (six and three times more, respectively, at 800 rotifers L$^{-1}$) than without prey. It was notable that *R. nomadica* exhibited increased swimming activity when fed in the rearing aquaria [5], suggesting that its metabolic demand under feeding conditions was higher. In addition, Moller and Riisgard [16] suggested that physiological processes, which are usually not considered in metabolic estimations, such as the mucus produced during predation and digestive leakage of dissolved organic carbon [69], could also affect the carbon budget. If so, compensating for the increased loss of carbon would require greater prey consumption, i.e., higher ingestion rates. It is also possible that not all of the ingested prey is entirely digested, as reported by Fraser [1], who found remains of copepod chitinous skeleton in *Sarsia princeps* (hydromedusa) fecal pellets. Similarly, Reeve et al. [70] found that at high prey density, the ctenophore *Mnemiopsis mccradyi* maintained a high feeding rate but released partially digested prey and replaced it with new ones. These findings also suggest that the minimum amount of prey required to compensate for the metabolic cost may be underestimated. Daan [71] found that at a constant low prey level of 50 copepods per liter, as in natural conditions where the study was conducted, 11 mm *Sarsia* hydromedusae reached maximum ingestion rates. Similarly, highest clearance rates were observed for small starved *M. mccradyi* at low prey concentrations of 0.5 copepods per liter [70]. In the same study, Reeve et al. [70] reported that continuous feeding reduced assimilation efficiency. Considering its branched feeding apparatus and strong swimming behavior, *R. nomadica* may best perform and achieve high carbon-specific ingestion rates in dense patches, which in its natural oligotrophic conditions, may be scarce and less extensive. We predicted *R. nomadica* would demonstrate a more efficient carbon budget, as evident by a shorter feeding duration required to satisfy its carbon demands. To test this prediction, we need to compare it with similar-sized individuals of related species, studied with an experimental protocol as close as possible to our own, a comparison as yet unavailable.

The results of the present study indicate that small medusae have higher mass-specific predation rates (on a per unit carbon basis) than larger medusae (Figure 7). When fed at prey concentrations of 100 L$^{-1}$, a 10 mm *R. nomadica* individual consumed per hour carbon equal to 46% of its body carbon content. In comparison, a 35 mm medusa at the same prey concentration consumed only 19% of its carbon mass, less than half of that consumed by the smaller medusa. Our observation of decreasing mass-specific ingestion rate with increasing size (=mass) of the medusae, corresponds with previous findings for many other pelagic predators [72], including gelatinous zooplankton [71,73]. In their synthesis on the mass scaling of respiration and maximum feeding and growth rates of different heterotrophic pelagic organisms, Kiorboe and Hirst [72] noted that not only mass-specific ingestion rates decrease with increasing organisms' size, but also growth and mortality rates decline with increasing body mass, suggesting they are adjusted to maximize the fitness of the organisms in their environment (i.e., reduce mortality and maintaining high feeding efficiency).

## 5. Summary

Since *R. nomadica* appears in the Israeli Mediterranean waters mainly in the summer months and often in a sexually mature state, we are unable to monitor its early life stages for growth and feeding preferences, hence, we are restricted to laboratory studies of these processes.

Nevertheless, the observed predation efficiency in relation to its metabolic demand may explain its large dimensions. Although feeding rates were obtained at high prey density, previous studies with ctenophores and hydromedusae suggest that predation and assimilation efficiencies are actually optimal at low prey density [40,71] and were even higher than those measured in the laboratory. These findings suggest that rhizostomes in general and *R. nomadica* in particular are specialized oligotrophic water feeders, efficiently locating scarce plankton patches, exploiting them to the full, maybe even at higher rates than we found. A lack of similar laboratory studies on other rhizostomes or even scyphomedusae, to which we could compare our results prevented us from strictly testing our working hypothesis and its derived predictions. However, the presented comparison with *A. aurita* is very suggestive of it being in the right direction and that filling this gap may indeed show *R. nomadica* to be a champion oligotrophic-water exploiter.

**Author Contributions:** Conceptualization, Z.K., D.K. and D.L.A.; methodology, Z.K., D.K. and D.L.A.; software, Z.K.; validation, Z.K., D.K. and D.L.A.; formal analysis, Z.K. and D.K.; investigation, Z.K.; resources, D.L.A.; data curation, Z.K.; writing—original draft preparation, Z.K.; writing—review and editing, D.L.A. and D.K.; visualization, Z.K.; supervision, D.L.A. and D.K.; project administration, D.L.A.; funding acquisition, D.L.A. All authors have read and agreed to the published version of the manuscript.

**Funding:** This project has received funding from the European Union's Horizon 2020 research and innovation programme under grant agreement No. 774499. The authors gratefully acknowledge support from the Maurice and Lady Hatter Fund of the Leon Recanati Institute for Maritime Studies (RIMS) at the University of Haifa.

**Institutional Review Board Statement:** Not applicable.

**Data Availability Statement:** Data is contained within the article.

**Acknowledgments:** This study would not have been possible without the tremendous help of the Maritime Aquaculture Department staff, Mevoot-Yam School, Michmoret: Rafi Yavetz, Gal Belogolovsky, Arik Weinberger, Ifat Mazoz and David Halfon. We would like to thank the School of Marine Sciences, Ruppin Academic Center, for providing marine laboratory facilities. We would like to gratefully thank Efrat Kessler, Emeritus, Goldshlager Eye Center-Sheba, Medicine-Sackler Faculty, Tel Aviv University, for kindly providing us with a bottle-roller which was essential for the laboratory feeding trials.

**Conflicts of Interest:** The authors declare no conflict of interest.

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
