# Peer review of "Respiration Rates, Metabolic Demands and Feeding of Ephyrae and Young Medusae of the Rhizostome Rhopilema nomadica"

_diversity, doi:10.3390/d13070320_

Round 1

Reviewer 1 Report

Review for the paper "Respiration rates, metabolic demands and feeding of the rhizostome

Rhopilema nomadica" by Zafrir Kuplik an co-authors submitted to "Diversity".

General comment.

The author provided experimental trials to respiratory and feeding rates in the rhizostome Rhopilema nomadica, a common scyphozoan in the eastern Mediterranean. They used standard methods of rearing, measuring, and statistical analysis. The authors showed differences in the parameters measured between different size classes and concluded that this species is a well-adapted predator which can significantly affect the zooplankton community in the area. Moreover, this study may be used for further research and monitoring of the Mediterranean Sea ecosystem.

Specific remarks.

Line 30. Consider replacing “in-situ” with “in situ”, “in situ” should be italicized

Line 33. Consider deleting “2009”

Line 35. Consider replacing “spatio-temporally” with “spatiotemporally”

Line 37. Consider deleting “1978”

Line 39. Consider replacing “large variability” with “high variability”

Line 48-49. Consider deleting “(e.g. Purcell 1992, Riisgard et al. 2012, Tilves et al. 2016”

Line 59. Consider replacing “BD” with “bell diameter (BD)”

Line 71. Consider replacing “size” with “the size”

Line 230. Consider replacing “decrease” with “a decrease”

Line 233. Consider replacing “number” with “the number”

Line 253. Consider replacing “log transformed” with “log-transformed”

Line 254. Consider replacing “log transformed” with “log-transformed”

Line 289-290. Consider replacing “log transformed” with “log-transformed”

Line 316. Consider replacing “total body carbon” with “the total body carbon”

Line 386. Consider replacing “in-situ” with “in situ”, “in situ” should be italicized

Line 389. Consider replacing “subsequently use” with “subsequent use”

Line 390. Consider replacing “The %” with “The proportion”

Line 413. Consider replacing “respiration rate” with “the respiration rate”

Line 414. Consider replacing “are best expressed” with “is best expressed”

Line 426. Consider replacing “mass specific” with “mass-specific”

Line 456. Consider replacing “=0.22” with “0.22”

Line 470. Consider replacing “is due to” with “are due to”

Line 509. Consider replacing “two-fold” with “twofold”

Line 516. Consider replacing “the only other” with “only one”

Line 552. Consider replacing “fast swimming” with “fast-swimming”

Line 582. Consider replacing “high feeding rate” with “a high feeding rate”

Line 625. Consider replacing “of R. nomadica’s” with “into R. nomadica’s”

References should be formatted according to Instructions for authors.

Author Response

Reviewer #1

General comment.

 The author provided experimental trials to respiratory and feeding rates in the rhizostome Rhopilema nomadica, a common scyphozoan in the eastern Mediterranean. They used standard methods of rearing, measuring, and statistical analysis. The authors showed differences in the parameters measured between different size classes and concluded that this species is a well-adapted predator which can significantly affect the zooplankton community in the area. Moreover, this study may be used for further research and monitoring of the Mediterranean Sea ecosystem.

 We greatly appreciate the positive feedback.

Specific remarks

All suggestions, excluding the two listed below, were accepted and the text was changed accordingly.

  • Line 233. Consider replacing “number” with “the number”

There is no need. The text already says "the remaining number…"

  • Line 516. Consider replacing “the only other” with “only one”

Since the previous sentence talks about missing data, changing the text to "only one" would be out of context.

Reviewer 2 Report

In the manuscript titled “Respiration rates, metabolic demands and feeding of the rhizostome Rhopilema nomadica” the authors present basic information on the feeding habits and energy demands of Rhopilema nomadica under relatively high prey densities in laboratory conditions. They find that Rhopilema nomadica is well adapted to patch-feeding in the hyperoligotrophic waters of the eastern Mediterranean Sea. Overall the manuscript is well written although some restructuring and additional information could help make the introduction and discussion more informative. Comments are included below:

  • The introduction presents relevant and important background but the authors should consider structuring it so that goals, specific endpoints, and hypotheses are clearly stated. Biometric relationships, respiration rates, carbon demands, ingestion and clearance rates and carbon budgets could all be introduced in the context of the study and presented in such a manner that the authors’ predictions are introduced within the framework of this organism being well adapted to its environment (patch-feeding, etc.) and then specifically addressed based on the results in the discussion.
  • The authors cite 84 references. This is on the high side and the authors could consider reducing the number by referencing reviews of the literature when possible.

Author Response

Reviewer #2

In the manuscript titled “Respiration rates, metabolic demands and feeding of the rhizostome Rhopilema nomadica” the authors present basic information on the feeding habits and energy demands of Rhopilema nomadica under relatively high prey densities in laboratory conditions. They find that Rhopilema nomadica is well adapted to patch-feeding in the hyperoligotrophic waters of the eastern Mediterranean Sea. Overall the manuscript is well written although some restructuring and additional information could help make the introduction and discussion more informative. Comments are included below:

  • The introduction presents relevant and important background but the authors should consider structuring it so that goals, specific endpoints, and hypotheses are clearly stated. Biometric relationships, respiration rates, carbon demands, ingestion and clearance rates and carbon budgets could all be introduced in the context of the study and presented in such a manner that the authors’ predictions are introduced within the framework of this organism being well adapted to its environment (patch-feeding, etc.) and then specifically addressed based on the results in the discussion.
  • The authors cite 84 references. This is on the high side and the authors could consider reducing the number by referencing reviews of the literature when possible

We highly appreciate the feedback and the constructive comments.

Starting with the last suggestion made – to reduce the number of references – we have reduced these, and now, rather than 84 we have 71 references.

Since a Major Revision was suggested, an extensive restructuring and refinement of the text were made.

Listed below are the reviewer's main suggestions and the corresponding text that served as the basis for the changes made throughout the entire manuscript.

As requested, the text was revised using Track Changes editing tool and resubmitted as such. In order to easily follow the changes that were made to the text and the corresponding line numbers, the text should be read with the No Markup option on.

  • …the authors should consider structuring it so that goals, specific endpoints, and hypotheses are clearly stated.

Introduction

Lines 47-58     "In view of this, our general working hypothesis was that it should possess relatively high metabolic efficiency and predation potential. This could be tested by evaluating several widely-used metabolic measures, also targeted by this study:

  1. Basal respiration rate - oxygen consumption (measured) and CO2 production (estimated), The prediction for these measures in nomadica is to be relatively low.
  2. Basal/minimum carbon demand, needed to cover the estimated CO2 production, accordingly also predicted to be low.
  3. Ingestion rate (measured) and carbon input (estimated), predicted to be high when compared to similarly-sized individuals of other species.
  4. Carbon budget – input versus required, when evaluated in terms of required feeding duration to satisfy daily demands, predicted to be relatively low in nomadica."

Lines 60-66     " In addition, we attempted to correlate different biometric features of the animals. Since respiration rates are commonly associated with biomass of the medusa, either wet/dry weight or carbon content [8,9], and since mass correlates with the bell diameter (BD) [10] it is suggested that only one of these measures is needed to estimate the others. Inferring both carbon content and respiration rates from BD will enable us to study the organisms in-situ and in the laboratory with minimal disruption and related artifacts. We will attempt to specifically address the above predictions when discussing our results."

  • Biometric relationships, respiration rates, carbon demands, ingestion and clearance rates and carbon budgets could all be introduced in the context of the study and presented in such a manner that the authors’ predictions are introduced within the framework of this organism being well adapted to its environment (patch-feeding, etc.) and then specifically addressed based on the results in the discussion.

With respect to the aforementioned suggestions made by the reviewer, we have referred to each of the specified targets of the study; basal respiration rate, basal/minimum carbon demand, ingestion rate, carbon budget and reflected on whether the results met our predictions or not, in the restructured Discussion.

The Summary section was also largely revised in order to address the newly defined hypotheses and our findings.

Round 2

Reviewer 2 Report

The authors of the manuscript titled “Respiration rates, metabolic demands and feeding of the rhizostome Rhopilema nomadica” have adequately addressed this reviewer’s comments and suggestions in the revised version of the manuscript.

Author Response

We thank the reviewer for their prior suggestion and for finding it satisfactory.